# SEMI-SUPERVISED DOMAIN ADAPTATION VIA JOINT ERROR BASED TRIPLET ALIGNMENT

## ABSTRACT

Existing domain adaptation methods are very effective in aligning feature distributions. However, these techniques usually do not improve the performance that much when a few annotated examples are available in the target domain. To address this semi-supervised domain adaptation scenario, we propose a novel joint error based triplet alignment approach that simultaneously optimizes the classification loss as well as the joint error among the source, labeled and unlabeled target domains. Besides, we propose a novel dissimilarity measurement between two classifiers, namely maximum cross margin discrepancy, which can asymptotically bridge the gap between the theory and algorithm. We empirically demonstrate the superiority of our method over several baselines.

## 1 INTRODUCTION

Given a large number of annotated training data, deep convolutional neural networks (Krizhevsky et al., 2012) are capable of significantly improving the performance of image classification, but usually cannot generalize well to new domains. Recent unsupervised domain adaptation (UDA) methods (Long et al., 2015; 2017; Ganin et al., 2016; Tzeng et al., 2017; Saito et al., 2017; Long et al., 2018) show effectiveness in adapting to unlabeled data from new domains by distribution alignment, but can fail to learn discriminative class boundaries especially when the domain gap is huge. Following Ben-David et al. (2010), most of the methods ignore the joint error and only focus on minimizing the source error and as well as the distance between domains. When aligning marginal distributions, samples from different classes can be grouped together if the domain gap is large enough (Fig.1b). In that case, the joint error becomes non-negligible and the target error cannot be properly bounded since no hypothesis can jointly classify the source and target data with a high accuracy (Ganin et al., 2016; Zhao et al., 2019). Zhang et al. (2023) provides a solution to this problem by incorporating the joint error into the target error upper bound in the unsupervised setting and we generalize this idea towards a Semi-Supervised Domain Adaptation (SSDA) setting where a few labeled target samples are available.

We propose a novel target error upper bound for SSDA that overcomes the limitations of previous methods and significantly improves the accuracy in new domains with only a few labeled target samples for each class. Our approach, namely Joint Error based Triplet Alignment (JTA), is based on simultaneously minimizing the joint error among different domains, as well as the error rate on labeled data, which can reduce the domain gap while learning discriminative features.

As shown in Fig.1c, based on the solution of UDA, simply treating labeled target data as a part of source data and performing the unsupervised framework to align the target distribution is not effective Saito et al. (2019). This is because simply training a model to jointly classify labeled target and source data does not necessarily merge the distributions in the feature space. If the source (red circle) and labeled target (green circle) domain are somehow far away, even if we reshape the unlabeled target (blue circle) domain to match the distributions in feature space, the classifier learned on the source and labeled target domain does not necessarily classify unlabeled target data. We believe matching distributions between the source and labeled target domains is as important as aligning the unlabeled target distribution. Therefore, this work aims to incorporate the alignment between the source and labeled target domains into the error bound of unlabeled target data.

According to the derivation of Ben-David et al. (2010), the triangle inequality is essential to build the target error bound. Besides, the measurement of the source error and terms related to marginal

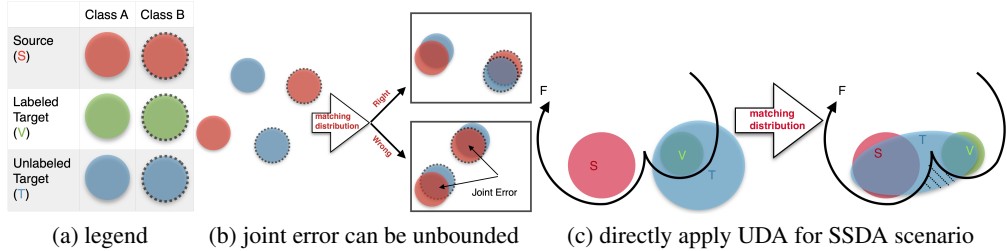

Figure 1: (a) Legend; (b) When domain gap is large, marginal distribution matching may lead to a wrong conditional alignment where the joint error (overlapping area) becomes non-negligible; (c) Feature distributions of the source and labeled target domains may not merge by purely optimizing the classification loss, which can harm the alignment performance for the unlabeled target domains (Arrow $F$ represents a hypothesis whose right side is classified as Class A assuming it points north).

discrepancy should be consistent. However, most of the methods that utilize the theory from Ben-David et al. (2010) somehow violate the above rules which is known as the gap between the algorithm and theory (Zhang et al., 2019). To address this problem, we propose a novel dissimilarity measurement between two hypotheses, namely maximum cross margin discrepancy (MCMD). Under a fair assumption, we can prove that the source error can be regarded as a special case of this measurement and the triangle inequality is asymptotically satisfied. Furthermore, when we apply MCMD to the proposed target error upper bound, we can prove that a part of the objective can be transformed into the CGAN objective (Mirza & Osindero, 2014) which is effective in aligning conditional distributions. Our contributions are summarized as follows:

- · We propose a novel target error upper bound, Joint Error based Triplet Alignment (JTA), which is designed for SSDA tasks;
- · We propose a novel dissimilarity measurement, namely maximum cross margin discrepancy (MCMD), which can asymptotically bridge the gap between the algorithm and theory;
- · We show our method's effectiveness on benchmark datasets for several SSDA tasks.

## 2 RELATED WORK

Semi-supervised domain adaptation (SSDA) is a crucial task (Saito et al., 2019; Yao et al., 2015; Ao et al., 2019; Donahue et al., 2013), however it has not been fully explored, especially when it comes to the learning theory. The main challenge is the gap between feature distributions of different domains, which can harm the source classifier's performance on target data. A typical solution to bridge the gap in unsupervised domain adaptation (UDA) is to distinguish source data from target data with a domain classifier which is fooled by a feature extractor (Ganin & Lempitsky, 2015; Tzeng et al., 2017; Long et al., 2018). Saito et al. (2017); Zhang & Harada (2019) argue that features obtained through a single domain classifier are not discriminative and try to minimize task-specific decision boundaries' disagreement on target data to push features far from decision boundaries. However, the alignment based on UDA methods does not necessarily match the feature space of source and labeled target domains. Saito et al. (2019) proposes a prototype network (Chen et al., 2018) based clusters that can leverage label target data and introduce a min-max game on the conditional entropy of unlabeled target data. Wang & Breckon (2020); Tang et al. (2020) share a similar strategy which can be described as cluster based iterative pseudo labeling. However, they assume target data is well clustered and can be labeled by euclidean distance from centroids, which may not hold true as high dimensional data can lie in a low dimensional manifold and suffer from the curse of dimensionality. In addition, the alignment based on clusters is usually not compact even if the centroids of each class from different domains are matched. Jiang et al. (2020) generates bidirectional adversarial samples from source to target domain and from target to source domain to fill the domain gap. Kim & Kim (2020) analyzes the target intra-domain discrepancy issue and suggests to minimize the gap using maximum mean discrepancy, perturbation loss, and class prototypes. Li et al. (2021) introduces an adversarial adaptive clustering loss to group features of unlabeled target data and performes feature alignment across domains. Yang et al. (2021) applies the co-training framework

where two classifiers corresponding to the decomposed semi-supervised learning (SSL) and UDA task exchange their confident predictions to iteratively teach each other. Singh (2021) employs class-wise contrastive learning and instance-level contrastive alignment to reduce the domain gap. Yoon et al. (2022) proposes a pair-based method that adapts a model to the target domain using self-distillation with sample pairs. Rahman et al. (2023) aligns domains by matching the bottleneck feature space of an auto-encoder. However, these methods lack a theoretical guarantee on the generalization error and the learning theory of SSDA is still left to be developed.

Methods based on generative (Dai et al., 2017; Salimans et al., 2016), ensemble (Laine & Aila, 2017), and adversarial approaches (Miyato et al., 2015) have improved the performance in SSL, but do not address the domain gap. Conditional entropy minimization is a widely used method in SSL (Erkan & Altun, 2010; Grandvalet & Bengio, 2005). However, Saito et al. (2019) shows that it fails to improve the performance when there is a large domain gap. Recent works (Li et al., 2021; Yang et al., 2021; Singh, 2021) combining powerful semi-supervised regularization (Sohn et al., 2020; Chen et al., 2020) and strong data augmentation (Cubuk et al., 2020) have achieved outstanding performance. However, these methods lack a theoretical analysis and do not work without the data augmentation. We propose a novel learning theory for SSDA that can take the advantage of UDA based alignment to address the domain gap as well as leverage the power of semi-supervised techniques to provide reliable decision boundaries on unlabeled target data.

## 3 PROPOSED METHOD

In this section, we first build a target error bound based on joint error for SSDA (Sec.3.1). This theoretical upper bound employs true labeling functions as the joint error would be intractable otherwise. Then, we introduce approximated labeling functions inside constrained hypothesis space to formalize an objective that can be optimized (Sec.3.3, 3.4, 3.5). Finally, we propose a novel discrepancy measurement to bridge the gap between the practical algorithm and theory (Sec.3.6).

### 3.1 JOINT ERROR BASED TRIPLET UPPER BOUND

In this section, we propose a joint error based upper bound of target error for SSDA. We consider the problem as a multi-class classification task where the learning algorithm has access to a set of $n$ labeled points $\{(x_s^i, y_s^i) \in (X \in \mathbb{R}^D \times Y = \{1, ..., K\})\}_{i=1}^n$ sampled i.i.d. from the source domain $S$, a set of $m$ unlabeled points $\{(x_t^i) \in X \in \mathbb{R}^D\}_{i=1}^m$ sampled i.i.d. from the unlabeled target domain $T$ and a set of $l$ labeled points $\{(x_v^i, y_v^i) \in (X \in \mathbb{R}^D \times Y = \{1, ..., K\})\}_{i=1}^l$ sampled i.i.d. from the labeled target domain $V$. Let $f_S : X \in \mathbb{R}^D \to \mathbb{R}^K$, $f_T : X \in \mathbb{R}^D \to \mathbb{R}^K$ and $f_V : X \in \mathbb{R}^D \to \mathbb{R}^K$ be the true labeling functions on the source, unlabeled and labeled target domains respectively, whose outputs are one-hot vectors denoting the corresponding labels of inputs. Let $\epsilon_D(f, f')$ denote a distance metric that measures the expectation of disagreement between two functions $f, f'$ over a distribution $D$. When we want to refer to the source error of a hypothesis $h \in H : X \in \mathbb{R}^D \to \mathbb{R}^K$, we use the shorthand $\epsilon_S(h) := \epsilon_S(h, f_S)$ that measures the disagreement w.r.t. the true labeling function $f_S$ over domain $S$. Similarly, we use $\epsilon_T(h)$, $\epsilon_V(h)$ to represent the error of the unlabeled and labeled target domains. With these notations, we propose the following upper bound for the target error[1]:

$$\epsilon_T(h) \le \frac{1}{2}[\epsilon_V(h) + \epsilon_S(h)] + D_{S,T,V}(f_S, f_T, f_V, h) = U(h) \tag{1}$$

$\frac{1}{2}[\epsilon_V(h) + \epsilon_S(h)]$ represents the **error rate on labeled data**. $D_{S,T,V}(f_S, f_T, f_v, h) = \frac{1}{2}[\epsilon_T(f_S, f_T) + \epsilon_T(f_V, f_T) + \epsilon_T(h, f_S) + \epsilon_T(h, f_V) + \epsilon_V(f_S, f_V) + \epsilon_S(f_V, f_S) - \epsilon_V(h, f_S) - \epsilon_S(h, f_V)]$ represents the **discrepancy among domains**.

Next we show the proposed upper bound is tightly related to the joint error as long as $S, V$ can be aligned. Owing to the triangle inequality, we further seek the upper bound and the lower bound of $U(h)$ and find that for any $h$, the following equations hold:

$$\begin{cases} U(h) & \le \frac{1}{2}[\epsilon_T(f_S, f_T) + \epsilon_T(f_V, f_T) + \epsilon_T(h, f_S) + \epsilon_T(h, f_V)] + \epsilon_V(f_S, f_V) + \epsilon_S(f_V, f_S) \\ U(h) & \ge \frac{1}{2}[\epsilon_T(f_S, f_T) + \epsilon_T(f_V, f_T) + \epsilon_T(h, f_S) + \epsilon_T(h, f_V)] \end{cases}$$

---

[1]See proof in D.1

Therefore the minimum of $U(h)$ is also bounded:

$$\begin{cases} \min_h U(h) & \leq \frac{1}{2}[\epsilon_T(f_S, f_T) + \epsilon_T(f_V, f_T) + \epsilon_T(f_V, f_S)] + \epsilon_V(f_S, f_V) + \epsilon_S(f_V, f_S) \\ \min_h U(h) & \geq \frac{1}{2}[\epsilon_T(f_S, f_T) + \epsilon_T(f_V, f_T) + \epsilon_T(f_V, f_S)] \end{cases}$$

Furthermore, we demonstrate the lower bound of $U(h)$ is equivalent to the upper bound of two optimal joint error terms $\lambda_{S,T}, \lambda_{V,T}$ where:

$$\epsilon_T(f_S, f_T) = \epsilon_T(f_S) + \epsilon_S(f_S) \geq \min_{h \in H}(\epsilon_T(h) + \epsilon_S(h)) = \lambda_{S,T}$$

in addition to the discrepancy between $f_S, f_V$ on $T$.

Now we can conclude that the minimum of $U(h)$ is lower bounded by the sum of two joint error terms ($\lambda_{S,T}, \lambda_{V,T}$) and the discrepancy between the source and labeled target domains. Meanwhile, it is also upper bounded by the above terms in addition to the joint error between source and labeled target domains ($\lambda_{S,V}$). Since $S, V$ are fully labeled, it is fair to assume their conditional distributions can be aligned, i.e. $f_S = f_V = f^\star$. And in that case, the minimum of $U(h)$ is achieved when $h = f^\star$, which is equivalent to the left-hand side of Eq.1 $:= \epsilon_T(f^\star, f_T) \geq \lambda_{S,V,T}$. This shows that the proposed upper bound is tightly related to the joint error.

### 3.2 INTUITION

In this section, we intuitively show how our upper bound is related to the alignment among domains. The minimum of $U(h)$ is an upper bound of 3 terms which are two joint error terms ($\epsilon_T(f_S, f_T), \epsilon_T(f_V, f_T)$) and the discrepancy between the source and labeled target domains ($\epsilon_T(f_V, f_S)$). During the learning process, our proposal can penalize the case where some samples from the source ($S$) and labeled target domains ($V$) are unmatched by reducing the discrepancy term $\epsilon_T(f_V, f_S)$, which is illustrated as the shadow area in Fig.2a. Besides, even if the centroids of each class from different domains are matched, our proposal can further stretch the feature space to make the alignment more compact by reducing the two joint error terms ($\epsilon_T(f_S, f_T), \epsilon_T(f_V, f_T)$) and the discrepancy ($\epsilon_T(f_V, f_S)$) which are illustrated as the shadow areas $1, 2, 1 + 2$ in Fig.2b respectively, such that the classifiers of the source ($S$) and labeled target domains ($V$) can provide more reliable predictions for the unlabeled target domain ($T$).

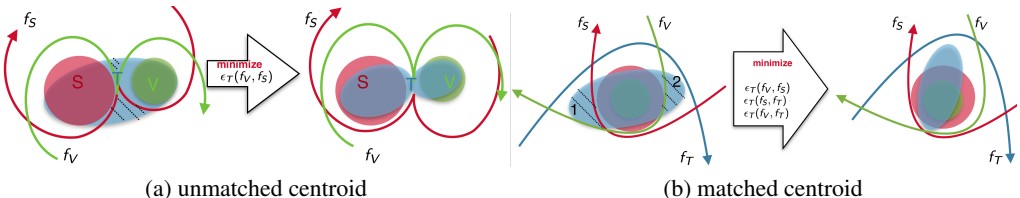

(a) unmatched centroid        (b) matched centroid

Figure 2: (a) Our proposal can reduce the shadow area which helps to penalize the case where the source and labeled target domains are unmatched; (b) Our proposal can reduce the shadow area which helps to further stretch the feature space and leads to a tighter alignment.

### 3.3 TRACTABILITY AND TIGHTNESS

In this section, we deal with the intractability of true labeling functions by introducing approximated labeling functions $f'_S, f'_T, f'_V$. If conditions: $f_S \in H_S \subseteq H, f_T \in H_T \subseteq H$ and $f_V \in H_V \subseteq H$ are met, the following holds:

$$D_{S,T,V}(f_S, f_T, f_V, h) \leq \max_{f'_S \in H_S, f'_T \in H_T, f'_V \in H_V} D_{S,T,V}(f'_S, f'_T, f'_V, h) \leq \max_{f'_S, f'_T, f'_V \in H} D_{S,T,V}(f'_S, f'_T, f'_V, h) \quad (2)$$

With the restriction on the size of the hypothesis space, we can build a tighter bound by taking supremum within hypothesis space $H_S, H_T, H_V$ instead of $H$:

$$\epsilon_T(h) \leq U(h) \leq \frac{1}{2}[\epsilon_S(h) + \epsilon_V(h)] + \max_{f'_S \in H_S, f'_T \in H_T, f'_V \in H_V} D_{S,T,V}(f'_S, f'_T, f'_V, h) \quad (3)$$

However, this raises two problems:

· **compatibility**: true labeling functions may not lie in any hypothesis space.

· **hypothesis space constraint**: how do we build hypothesis spaces that satisfy the conditions.

We will deal with these problems respectively in the following sections.

### 3.4 COMPATIBILITY

In this section, we show there exist replacements of true labeling functions inside $H$. For any $f_S^*, f_T^*, f_V^* \in H$, the following equation holds[2]:

$$\epsilon_T(h) \leq \frac{1}{2}[\epsilon_V(h) + \epsilon_S(h)] + D_{S,T,V}(f_S^*, f_T^*, f_V^*, h) + \theta \tag{4}$$

Therefore Eq.1 can be rewritten as a similar upper bound where $f_S, f_T, f_V$ are replaced with $f_S^*, f_T^*, f_V^*$ in addition to a residual term $\theta$:

$$\theta = \frac{1}{2}\epsilon_S(f_S, f_S^*) + \epsilon_V(f_S, f_S^*) + \epsilon_T(f_S, f_S^*) + \frac{1}{2}\epsilon_V(f_V, f_V^*) + \epsilon_S(f_V, f_V^*) + \epsilon_T(f_V, f_V^*) + \epsilon_T(f_T, f_T^*) \tag{5}$$

As long as the algorithm is run within finite samples, there exist $f_S^*, f_T^*, f_V^*$ inside $H$ with enough complexity that can lead to a residual $\theta$ very close to zero thus showing that true labeling functions can be replaced with $f_S^*, f_T^*, f_V^*$.

### 3.5 HYPOTHESIS SPACE CONSTRAINT

In this section, we show how to build constrained hypothesis spaces that lead to a tighter bound. As proved in Sec.3.4, we can replace true labeling functions $f_S, f_T, f_V$ with $f_S^*, f_T^*, f_V^* \in H$ to maintain the upper bound even if true labeling functions do not lie in $H$. Based on the argument in Eq.2, in order to obtain a tighter upper bound, we need to further construct hypothesis spaces that include these replacements: $f_S^* \in H_S, f_T^* \in H_T, f_V^* \in H_V$.

$\theta$ in Eq.5 gives us clues for the constraints of hypothesis spaces, which implies $f_S^*, f_V^*$ must be consistent with $f_S, f_V$ on $S, V, T$ and $f_T^*$ has to follow $f_T$ on $T$. But in practice, we cannot track the behavior of true labeling functions on the unlabeled target domain $T$. Therefore, we ignore the intractable constraints and build a relaxed space to make sure that space contains those replacement functions $f_S^*, f_T^*, f_V^*$. For instance, if we can build a space $H_S$ that includes all hypotheses consistent with $f_S$ on $S, V$, then this space must contain $f_S^*$ since it lies in a subspace of $H_S$. It is fair to assume $f_S, f_V$ can both classify $V, S$ since the two domains can be perfectly aligned given true labels. Then, according to $\theta$, the hypothesis spaces $H_S, H_T, H_V$ are given by:

$$\begin{cases} H_S = \{f_S' | \arg\min_{f_S' \in H}[\frac{1}{2}\epsilon_S(f_S') + \epsilon_V(f_S')]\} \\ H_T = \{f_T' | \arg\min_{f_T' \in H} \epsilon_T(f_T')\} \\ H_V = \{f_V' | \arg\min_{f_V' \in H}[\frac{1}{2}\epsilon_V(f_V') + \epsilon_S(f_V')]\} \end{cases} \tag{6}$$

In order to build a reliable hypothesis space for $H_T$ which lacks information, we approximate the target error with the error rate on labeled data and a semi-supervised regularization term:

$$\epsilon_T(f_T') \approx \frac{1}{2}[\epsilon_S(f_T') + \epsilon_V(f_T')] + L_{reg} \tag{7}$$

where $L_{reg}$ represents any semi-supervised regularization that helps to find a better classifier given unlabeled target data. Details of $L_{reg}$ are described in Sec.3.7.

### 3.6 MAXIMUM CROSS MARGIN DISCREPANCY

In this section, we propose a novel discrepancy measurement for the loss function $\epsilon$ to bridge the gap between theory and algorithm, which is usually caused by inconsistent choices of loss functions for source error and discrepancy as well as the violation of triangle inequality. Following the above notations, we consider a hypothesis $h \in H : X \in \mathbb{R}^D \to \mathbb{R}^K$ (e.g. a neural network whose output

---

[2]See proof in D.2

layer is a softmax function) for multi-class classification where $h(y|x)$ indicates y-th elements of the output given input $x$. Thus an induced labeling function named $l_h$ from $X \rightarrow Y$ is given by:

$$l_h : x \rightarrow \arg\max_{y \in Y} h(y|x)$$

Inspired by maximum-margin classifier (Koltchinskii & Panchenko, 2002) and GAN (Goodfellow et al., 2014), we give the definition for the maximum cross margin discrepancy (MCMD) for $h_1, h_2 \in H$ over $D$ as follows:

$$\epsilon_D(h_1, h_2) = \mathbb{E}_{x \in D}[\text{mcmd}(h_1, h_2; x)]$$

Considering the discrepancy between two hypotheses $h_1$ and $h_2$ where $y_1 = l_{h_1}(x), l_{h_2}(x) = y_2$:

$$\text{mcmd}(h_1, h_2; x) = \max(|\log h_1(y_1|x) - \log h_2(y_1|x)|, |\log h_2(y_2|x) - \log h_1(y_2|x)|) \quad (8)$$

The proposed MCMD satisfies the triangle inequality that can bridge the gap between the theory and actual algorithm (proof in D.3). Besides, it can be interpreted in terms of CGAN(Mirza & Osindero, 2014) that can theoretically guarantee the conditional distribution alignment (proof in D.4).

Following the trick introduced by Goodfellow et al. (2014) to mitigate the burden of exploding or vanishing gradients, we optimize $\log(1 - h(y|x))$ instead of $-\log h(y|x)$ in practice.

### 3.7 TRAINING OBJECTIVE

In this section, we first introduce several commonly used semi-supervised regularization that can help to approximate the target error. Then we describe the overall loss function based on MCMD. We introduce a feature extractor $g : X \in \mathbb{R}^D \rightarrow \mathbb{R}^F$ that can map original inputs into feature space: $S_g = \{(g(x), y)|(x, y) \sim S\}, V_g = \{(g(x), y)|(x, y) \sim V\}, T_g = \{g(x)|x \sim T\}$ as well as hypotheses $h, f'_S, f'_T, f'_V \in H^F : \mathbb{R}^F \rightarrow \mathbb{R}^K$.

#### 3.7.1 REGULARIZED ENTROPY MINIMIZATION

Entropy minimization (Grandvalet & Bengio, 2005) adds a loss term that encourages the network to make confident (low-entropy) predictions for all unlabeled examples regardless of their class, which can push the classifier to be more discriminative. The second term impose a class balance prior to penalize conditional models with complex decision boundaries in order to yield sensible solutions (Tang et al., 2020; Gomes et al., 2010; Saito et al., 2017).

$$L_{ent} = -\mathbb{E}_{x \in T} \sum_k f'_T(y = k|g(x)) \log f'_T(y = k|g(x)) + \sum_k \mathbb{E}_{x \in T}[f'_T(y = k|g(x))] \log \mathbb{E}_{x \in T_g}[f'_T(y = k|g(x))]$$

#### 3.7.2 PSEUDO LABELING

Pseudo labeling is a classic method for semi-supervised learning. Here, we choose the progressive pseudo labeling technique introduced in Tang et al. (2020); Sohn et al. (2020). For the same input with random augmentations $x, x' \in T$, we minimize the cross entropy loss for $x$ using pseudo labels given by the predictions larger that a threshold $\rho$ on $x'$, where $\mathbb{1}(\cdot)$ is the indication function.

$$L_{pse} = -\mathbb{E}_{x, x' \in T} \mathbb{1}(\max_{y'} h(y'|g(x')) > \rho) \log f'_T(y'|g(x))$$

#### 3.7.3 CONSISTENCY REGULARIZATION

$\Pi$-Model (Laine & Aila, 2017; Sajjadi et al., 2016) adds a loss term which encourages the distance between a network's output for the same input with random augmentations $x, x' \in T$ to be small.

$$L_{con} = \mathbb{E}_{x, x' \in T}|f'_T(g(x)) - f'_T(g(x'))|$$

Given hyper-parameters $\lambda_e, \lambda_c, \rho$, the semi-supervised regularization loss can be written as:

$$L_{reg} = \lambda_e L_{ent} + \lambda_c L_{con} + L_{pse}$$

### 3.7.4 Overall Loss

Firstly, we formalize the **error rate on labeled data** in the upper bound (Eq.3):

$$\boldsymbol{L_{ce}} = \frac{1}{2}[\epsilon_{S_g}(h) + \epsilon_{V_g}(h)] \tag{9}$$

We define the source error of a hypothesis $h$ based on MCMD (Eq.8):

$$\epsilon_{S_g}(h) = \epsilon_{S_g}(h, f_S) = \mathbb{E}_{x,y \in S}[\text{mcmd}(h, f_S; g(x))] = \mathbb{E}_{x,y \in S}|\log f_S(y|x) - \log h(y|g(x))| = -\mathbb{E}_{x,y \in S} \log h(y|g(x))$$

where the source error can be expressed as a cross entropy loss since true labeling functions map the inputs into one-hot vectors denoting their corresponding labels. $\epsilon_{V_g}(h)$ can be defined analogously.

Secondly, we formalize the **discrepancy among domains** in the upper bound (Eq.3) based on MCMD:

$$
\begin{aligned}
\boldsymbol{L_{dis}} &= D_{S_g, T_g, V_g}(f'_S, f'_T, f'_V, h) \\
&= \frac{1}{2}\{\mathbb{E}_{x \in T}[\text{mcmd}(f'_S, f'_T; g(x)) + \text{mcmd}(f'_T, f'_V; g(x)) + \text{mcmd}(f'_S, h; g(x)) + \text{mcmd}(f'_V, h; g(x))] \\
&\quad + \mathbb{E}_{x \in S}[\text{mcmd}(f'_S, f'_V; g(x)) - \text{mcmd}(f'_V, h; g(x))] + \mathbb{E}_{x \in V}[\text{mcmd}(f'_S, f'_V; g(x)) - \text{mcmd}(f'_S, h; g(x))]\}
\end{aligned}
\tag{10}
$$

Then, we formalize **constraints** (Eq.6,7 in Sec.3.5) for approximated labeling functions $f'_S, f'_T, f'_V$ to make sure they lie in the proper hypothesis spaces $H_S, H_T, H_V$ according to Eq.3:

$$
\begin{cases}
\boldsymbol{L_{H_S}} = \frac{1}{2}\epsilon_{S_g}(f'_S) + \epsilon_{V_g}(f'_S) \\
\boldsymbol{L_{H_T}} = \frac{1}{2}[\epsilon_{S_g}(f'_T) + \epsilon_{V_g}(f'_T)] + L_{reg} \\
\boldsymbol{L_{H_V}} = \frac{1}{2}\epsilon_{V_g}(f'_V) + \epsilon_{S_g}(f'_V)
\end{cases}
\tag{11}
$$

Finally, by introducing a trade-off parameter $\lambda$ to balance the classification loss and discrepancy, the overall objective function of the upper bound (Eq.3) can be written as:

$$
\begin{cases}
\min_{f'_S, f'_T, f'_V \in H^F} L_{H_S} + L_{H_T} + L_{H_V} - \lambda L_{dis} \\
\min_{h \in H^F, g} L_{H_S} + L_{H_T} + L_{H_V} + L_{ce} + \lambda L_{dis}
\end{cases}
$$

## 4 Evaluation

### 4.1 Datasets and Implementation

We evaluate our proposal on several popular benchmark datasets, including VisDA2017 (Peng et al., 2017), DomainNet (Peng et al., 2019), and Office-Home (Venkateswara et al., 2017). DomainNet is a recent benchmark dataset for large-scale domain adaptation that has 345 classes and 6 domains. Following the protocol established in Saito et al. (2019), we pick 4 domains (Real, Clipart, Painting, Sketch) with 126 classes for the experiments. VisDA2017 is a synthetic-to-real domain adaptation benchmark, which consists of 150k synthetic and 55k real images from 12 categories. Office-Home consists of 4 domains (Real, Clipart, Product, Art) and 65 categories. We evaluate our method by fine-tuning a ResNet-34 (He et al., 2015) model pretrained on ImageNet (Deng et al., 2009). We introduce RandAugment (Cubuk et al., 2020) other than commonly used RandomFlip and RandomCrop to make a fair comparison with recent methods(Li et al., 2021; Yang et al., 2021; Singh, 2021) that use strong data augmentation. Hyper-parameters are set to $\lambda = 0.01, \lambda_e = 0.5, \lambda_c = 30, \rho = 0.8$ in all benchmarks (see A for implementation, B for hyper-parameter selection, C.2 for ablation study).

### 4.2 Results

**DomainNet** We evaluate our proposal w/ and w/o RandAugment to provide a fair comparison with previous SSDA methods STar, ATDOC, APE, BiAT, MJE, MME. As shown in Tab.1, our proposal provides a reliable performance in both conditions. Under this fair comparison, our method outperforms STar by 0.8% and CDAC by 0.5% on average. We also show that recent algorithms like CDAC heavily depend on strong data augmentation and without it, those algorithms may not outperform previous methods like STar.

Table 1: Accuracy (%) on DomainNet under the setting of 3-shot using ResNet34. ⋆ denotes the methods w/o additional data augmentation other than RandomFlip and RandomCrop.

| METHOD | R to C | R to P | P to C | C to S | S to P | R to S | P to R | MEAN |
|---|---|---|---|---|---|---|---|---|
| S+V⋆ | 60.0 | 62.2 | 59.4 | 55.0 | 59.5 | 50.1 | 73.9 | 60.0 |
| DANN⋆ (Ganin et al., 2016) | 59.8 | 62.8 | 59.6 | 55.4 | 59.9 | 54.9 | 72.2 | 60.7 |
| CDAN⋆ (Long et al., 2018) | 69.0 | 67.3 | 68.4 | 57.8 | 65.3 | 59.0 | 78.5 | 66.5 |
| ENT⋆ (Grandvalet & Bengio, 2005) | 71.0 | 69.2 | 71.1 | 60.0 | 62.1 | 61.1 | 78.6 | 67.6 |
| MME⋆ (Saito et al., 2019) | 72.2 | 69.7 | 71.7 | 61.8 | 66.8 | 61.9 | 78.5 | 68.9 |
| MJE⋆ (Zhang & Harada, 2019) | 74.7 | 71.3 | 74.6 | 62.3 | 67.4 | 63.9 | 79.3 | 70.0 |
| BiAT⋆ (Jiang et al., 2020) | 74.9 | 68.8 | 74.6 | 61.5 | 67.5 | 62.1 | 78.6 | 69.7 |
| APE⋆ (Kim & Kim, 2020) | 76.6 | 72.1 | 76.7 | 63.1 | 66.1 | 67.8 | 79.4 | 71.7 |
| ATDOC⋆(Liang et al., 2021) | 76.9 | 72.5 | 74.2 | 66.7 | 70.8 | 64.6 | 81.2 | 72.4 |
| STar⋆(Singh et al., 2021) | 77.1 | 73.2 | 75.8 | 67.8 | 69.2 | 67.9 | 81.2 | 73.2 |
| CDAC(Li et al., 2021) | 79.6 | 75.1 | **79.3** | 69.9 | 73.4 | 72.5 | 81.9 | 76.0 |
| DECOTA(Yang et al., 2021) | **80.4** | 75.2 | 78.7 | 68.6 | 72.7 | 71.9 | 81.5 | 75.6 |
| CLDA(Singh, 2021) | 77.7 | 75.7 | 76.4 | 69.7 | **73.7** | 71.1 | 82.9 | 75.3 |
| S³D⋆(Yoon et al., 2022) | 75.9 | 72.1 | 75.1 | 64.4 | 70.0 | 66.7 | 80.3 | 72.1 |
| AESL⋆(Rahman et al., 2023) | 77.3 | 74.1 | 76.2 | 65.2 | 69.6 | 69.5 | 80.5 | 73.2 |
| CDAC⋆ | 75.3 | 72.6 | 74.7 | 65.9 | 70.6 | 67.7 | 80.2 | 72.4 |
| Ours⋆ | 77.6 | 73.8 | 76.5 | 67.6 | 71.2 | 69.5 | 81.6 | 74.0 |
| Ours | 80.1 | **75.7** | 79.1 | **70.7** | 73.1 | **73** | **83.5** | **76.5** |

Table 2: Accuracy (%) on Office-Home & ViSDA under the setting of 3-shot using ResNet34. ⋆ denotes the methods w/o additional data augmentation other than RandomFlip and RandomCrop.

| METHOD | R→C | R→P | R→A | P→R | P→C | P→A | A→P | A→C | A→R | C→R | C→A | C→P | MEAN | VisDA (Avg) |
|---|---|---|---|---|---|---|---|---|---|---|---|---|---|---|
| S+V⋆ | 55.7 | 80.8 | 67.8 | 73.1 | 53.8 | 63.5 | 73.1 | 54.0 | 74.2 | 68.3 | 57.6 | 72.3 | 66.2 | 67.4 |
| DANN⋆ | 57.3 | 75.5 | 65.2 | 69.2 | 51.8 | 56.6 | 68.3 | 54.7 | 73.8 | 67.1 | 55.1 | 67.5 | 63.5 | 70.1 |
| ENT⋆ | 62.6 | 85.7 | 70.2 | 79.9 | 60.5 | 63.9 | 79.5 | 61.3 | 79.1 | 76.4 | 64.7 | 79.1 | 71.9 | 73.7 |
| MME⋆ | 64.6 | 85.5 | 71.3 | 80.1 | 64.6 | 65.5 | 79.0 | 63.6 | 79.7 | 76.6 | 67.2 | 79.3 | 73.1 | 76.6 |
| MJE⋆ | 65.1 | 85.6 | 74.7 | 80.4 | 62.7 | 66.5 | 78.8 | 63.9 | 80.3 | 76.8 | 66.4 | 78.7 | 73.3 | 80.3 |
| APE⋆ | 66.4 | 86.2 | 73.4 | 82.0 | 65.2 | 66.1 | 81.1 | 63.9 | 80.2 | 76.8 | 70.0 | 74.0 | 74.0 | 80.5 |
| CDAC | 67.8 | 85.6 | 72.2 | 81.9 | 67.0 | 67.5 | 80.3 | 65.9 | 80.6 | 80.2 | 67.4 | 81.4 | 74.2 | 79.0 |
| DECOTA | 70.4 | **87.7** | 74.0 | 82.1 | 68.0 | 69.9 | 81.8 | 64.0 | 80.5 | 79.0 | 68.0 | 83.2 | 75.7 | 79.5 |
| CLDA | 66.0 | 87.6 | **76.7** | 82.2 | 63.9 | **72.4** | 81.4 | 63.4 | 81.3 | 80.3 | 70.5 | 80.9 | 75.5 | - |
| Ours⋆ | 67.3 | 86.7 | 72.5 | 81.8 | 66.1 | 68.7 | 81.6 | 64.9 | 80.7 | 80.0 | 70.1 | 83.0 | 75.3 | 84.7 |
| Ours | **70.6** | 86.4 | 73.9 | **82.8** | **68** | 69.9 | **83.1** | **67.1** | **82** | **80.9** | 70.6 | **83.3** | **76.6** | **87.1** |

**Office-Home**   As shown in Tab.2, results on the Office-Home dataset further validate the effectiveness of our proposal which gives a performance very close to recent methods without strong data augmentation. Data augmentation is not necessary for our proposal since the target error is theoretically bounded, unlike CDAC, DECOTA and CLDA where the strong data augmentation is an essential built-in operation.

**VisDA**   As shown in Tab.2, our proposal outperforms others by a large margin even without strong data augmentation. We fine-tune the hyper-parameters for CDAC and DECOTA to obtain a decent result since the original is worse than $S + V$ (a model purely trained on labeled data).

## 4.3 ANALYSIS

### 4.3.1 FEATURE VISUALIZATION

We plot learned features of Real to Clipart task from DomainNet with t-SNE (van der Maaten & Hinton, 2008) in Fig.3. Fig.3d shows features of unlabeled target data, where each color represents a different class. In our method, most of the target samples are well-clustered and do not have a large variance within the class. In Fig.3h, our method almost perfectly matches conditional distributions of the two feature spaces as we expect. We also plot features of the source (red) and target domains (blue) in Fig.3l to show that our proposal can align marginal distributions. Besides, in our method, each cluster is clearly separated while others sometimes merge different clusters.

### 4.3.2 QUANTITATIVE FEATURE ANALYSIS

We quantitatively investigate the characteristics of the extracted features of Real to Clipart task from DomainNet for different methods. In Fig.4b we plot the ratio between inter-cluster distance and intra-cluster distance for each dimension of the extracted features from target domain in descending order. A larger value indicates a more discriminative feature according to the discriminant analysis (Fisher, 1936). In Fig.4c, we calculate A-distance by training a SVM (Vapnik & Lerner, 1963) based domain classifier as proposed in Ben-David et al. (2007). Our method greatly reduces the domain

divergence compared to other methods. Fig.4a shows that the learning process is stable and all classifiers will finally reach a reliable convergence.

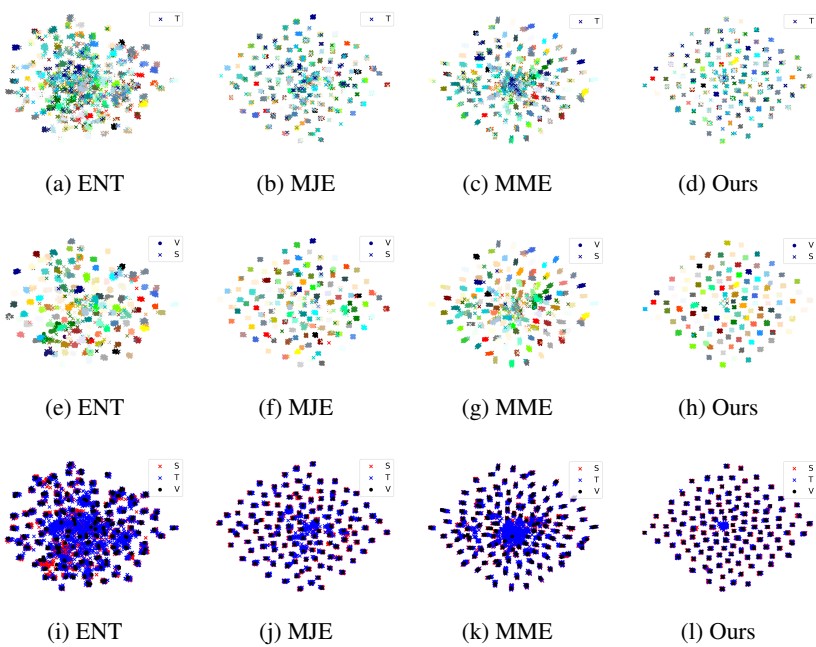

| (a) ENT | (b) MJE | (c) MME | (d) Ours |

| (e) ENT | (f) MJE | (g) MME | (h) Ours |

| (i) ENT | (j) MJE | (k) MME | (l) Ours |

Figure 3: Comparisons of the feature space visualized by t-SNE after the adaptation from Real to Clipart; (a)-(d) show feature spaces of unlabeled target domain, and our method gives better alignment where each cluster is clearly separated and most of the samples from the same class are grouped together; (e)-(h) show the alignment between source and labeled target domains where ours achieves a perfect conditional distribution alignment; (i)-(l) show the alignment between marginal distributions where ours gives a tighter match.

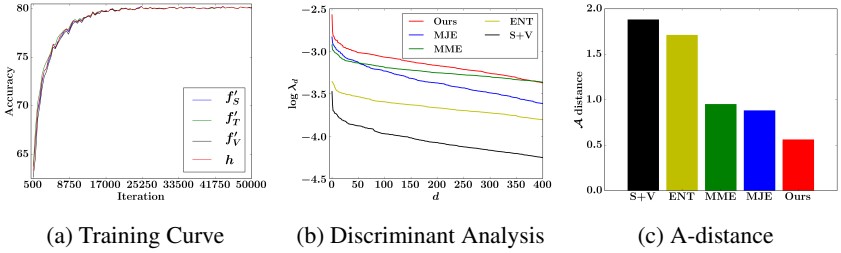

(a) Training Curve     (b) Discriminant Analysis     (c) A-distance

Figure 4: (a) Training procedure is stable and all the classifiers will finally reach a reliable convergence; (b) Ratio between inter-cluster distance and intra-cluster distance for each dimension of the extracted features from target domain. A high ratio in our method means the extracted features are more discriminative; (c) Our method clearly reduces the domain divergence.

## 5 CONCLUSION

We propose a novel Joint Error based Triplet Alignment approach that adversarially optimizes an upper bound of the joint error for semi-supervised domain adaptation, and a novel Maximum Cross Margin Discrepancy for dissimilarity measurement that asymptotically bridges the gap between the algorithm and theory. Adaptation is achieved by jointly aligning the conditional distributions among different domains as well as minimizing the error rate on labeled data. We empirically demonstrated the superiority of our method over many baselines.

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

---

**Algorithm 1** JTA

---

**Input**: source data $S$, labeled target data$V$, unlabeled target data $T$
**Parameter**: approximated labeling functions $f'_S, f'_T, f'_V \in H^F : \mathbb{R}^F \to \mathbb{R}^K$; feature extractor $g : X \in \mathbb{R}^D \to \mathbb{R}^F$; hypothesis $h \in H^F : \mathbb{R}^F \to \mathbb{R}^K$;
**Hyper-parameter**:trade-off parameter $\lambda$; learning rate $\alpha$
**Output**: updated parameters $w = (g, h, f'_S, f'_T, f'_V)$
**Notation**: gradient reversal operator $R(\cdot)$; reversed feature space $S^R_g = \{R(g(x)), y | x, y \in S\}, V^R_g = \{R(g(x)), y | x, y \in V\}, T^R_g = \{R(g(x)) | x \in T\}$; reversed hypothesis $h^R : z \in \mathbb{R}^F \to R(h(R(z))) \in \mathbb{R}^K$;
**for** $iteration = 1, 2, \ldots$ **do**
    **Step A**: compute classification loss on labeled data $L_{ce}$ (Eq.9) and hypothesis constraints $L_{H_S}, L_{H_T}, L_{H_V}$ (Eq.11)
    **Step B**: compute discrepancy $L^R_{dis}$ (Eq.10) given the gradient reversal layer:
        $L^R_{dis} = D_{S^R_g, T^R_g, V^R_g}(f'_S, f'_T, f'_V, h^R)$
    **Step C**: minimize overall objective w.r.t. all parameters $w$
      Update $w$:
        $w \leftarrow w + \alpha \Delta w$ where $\Delta w = -\frac{\partial(L_{H_S} + L_{H_T} + L_{H_V} + L_{ce} - \lambda L^R_{dis})}{\partial w}$
**end for**

---

# A    DETAILS OF IMPLEMENTATION

For a fair comparison, we use exactly the same labeled source data, labeled target data and unlabeled target data as Saito et al. (2019). We provide details of our implementation in Alg.1, where we introduce a gradient reversal layer (Ganin et al., 2016) to train the overall objective all together. Note that we do not optimize $f'_T$ on semi-supervised regularization losses in practice as it can lead to an early convergence to bad local optimum. The pre-trained model (e.g., ResNet34) except the last layer combined with a single-layer bottleneck (Zhang et al., 2019) is used as feature extractor $g$ and randomly initialized 2-layer fully-connected networks are used for classifiers $f'_S, f'_T, f'_V, h$. We introduce spectral normalization (Miyato et al., 2018) to ensure the classifiers are approximately Lipschitz which makes the adversarial learning more stable. We further utilize the smoothed cross-entropy loss (Müller et al., 2019; Liang et al., 2020) to prevent the network from becoming over-confident on labeled data. We adopt SGD with momentum 0.9 for optimization where the learning rate is set to $\alpha$ for all fully-connected layers whereas it is set to $0.1\alpha$ for the other convolution layers. The initial learning rate is set to 0.01 for DomainNet, 0.004 for Office-Home, 0.001 for VisDA according to Zhang et al. (2019); Saito et al. (2019); Zhang & Harada (2019). We employ learning rate annealing strategy proposed in Ganin et al. (2016). We use RandomFlip, RandomCrop and RandAugment as data augmentation and the batch size is fixed to 32. The results of adaptation scenarios from all three benchmarks DomainNet, Office-Home, VisDA are given by 50k iterations run on Tesla V100.

# B    HYPER-PARAMETER SELECTION

We set $\lambda = 0.01, \lambda_c = 30$ in all benchmarks according to Zhang & Harada (2019); Yang et al. (2021). We fine-tune $\lambda_e, \rho$ to obtain a better performance based on a validation set containing 3 labeled target samples per class from DomainNet dataset C to S scenario. In Fig.5. , we show the performance when varying the hyper-parameters $\lambda_e, \rho$. Given the validation accuracy, we set $\lambda_e = 0.5, \rho = 0.8$ in all benchmarks.

# C    ADDITIONAL EXPERIMENTS

## C.1    VARYING NUMBER OF LABELED EXAMPLES

Fig.6 shows the behavior of different methods when the number of labeled examples in the target domain varies from 0 to 20 per class on DomainNet using ResNet34 backbone. Cluster based methods like MME (Saito et al., 2019) will finally be outperformed by a simple entropy minimization when

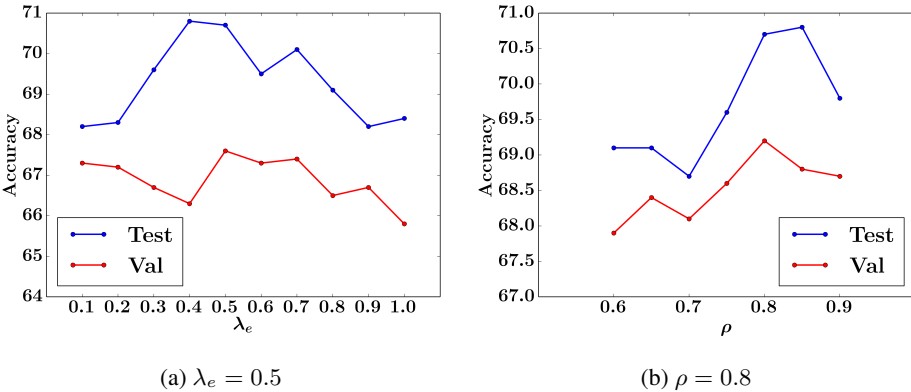

(a) $\lambda_e = 0.5$ (b) $\rho = 0.8$

Figure 5: Sensitivity w.r.t hyper-parameters $\lambda_e, \rho$ tested on C→S scenario in DomainNet. The hyper-parameters are set to the same values for all benchmarks based on the validation accuracy .

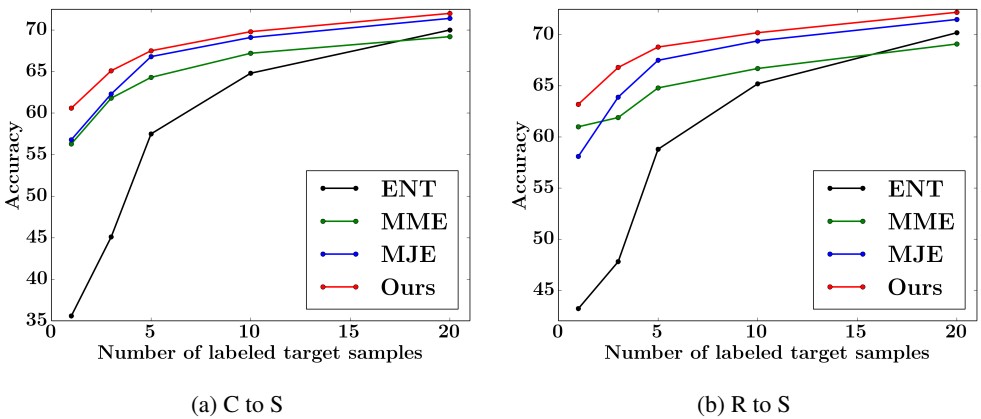

(a) C to S (b) R to S

Figure 6: Accuracy vs the number of labeled target samples on DomainNet using ResNet34 backbone. Our method maintains high level performance for different sample size of the labeled target domain.

sample size grows. On the contrary our method maintains a high level performance for various size of the labeled target data.

## C.2 ABLATION STUDY

We conduct ablation studies on the Office-Home of A → R and C → P under 3-shot setting, as shown in Tab.3. Our proposal outperforms UDA based alignment approach DANN by 4.9% and 12.5% respectively without any semi-supervised regularization or strong data augmentation, which demonstrates the effectiveness of our learning theory for SSDA. Among the semi-supervised regularization, entropy minimization contributes most to the performance gain as it leads to a more discriminative $f'_T$ on $T$, which is required by the hypothesis space constraint. Without strong data augmentation, FixMatch (Sohn et al., 2020) based pseudo labeling does not yield much improvement. As a different type of semi-supervised regularization, consistency loss can usually slightly boost the performance.

Table 3: Ablation studies of semi-supervised regularization losses. We report the Accuracy (%) on Office-Home of A → R and C → P under the setting of 3-shot using a ResNet34 backbone. $\star$ denotes the methods w/o additional data augmentation other than RandomFlip and RandomCrop.

| METHOD | $L_{ent}$ | $L_{pse}$ | $L_{con}$ | A → R | C → P |
|--------|-----------|-----------|-----------|-------|-------|
| S+V$\star$ | | | | 74.3 | 72.3 |
| DANN$\star$ | | | | 73.8 | 67.5 |
| | | | | 78.7 | 80.0 |
| Ours$\star$ | ✓ | | | 80.1 | 82.5 |
| | ✓ | ✓ | | 80.5 | 82.8 |
| | ✓ | ✓ | ✓ | 80.7 | 83.0 |

# D    PROOF

## D.1    PROOF OF EQ.1

$$
\begin{aligned}
\epsilon_T(h) &= \frac{1}{2}[2\epsilon_T(h, f_T) - \epsilon_V(h, f_V) - \epsilon_S(h, f_S) + \epsilon_V(h, f_V) + \epsilon_S(h, f_S) \\
&\quad + \epsilon_T(h, f_S) + \epsilon_T(h, f_V) - \epsilon_T(h, f_S) - \epsilon_T(h, f_V) + \epsilon_V(h, f_S) + \epsilon_S(h, f_V) - \epsilon_V(h, f_S) - \epsilon_S(h, f_V)] \\
&= \frac{1}{2}([\epsilon_T(h, f_T) - \epsilon_T(h, f_S)] + [\epsilon_T(h, f_T) - \epsilon_T(h, f_V)] + \epsilon_T(h, f_S) + \epsilon_T(h, f_V) \\
&\quad + [\epsilon_V(h, f_S) - \epsilon_V(h, f_V)] + [\epsilon_S(h, f_V) - \epsilon_S(h, f_S)] - \epsilon_V(h, f_S) - \epsilon_S(h, f_V) + \epsilon_V(h, f_V) + \epsilon_S(h, f_S)) \\
&\leq \frac{1}{2}([\epsilon_T(f_S, f_T) + \epsilon_T(f_V, f_T) + \epsilon_T(h, f_S) + \epsilon_T(h, f_V) + \epsilon_V(f_S, f_V) + \epsilon_S(f_V, f_S) \\
&\quad - \epsilon_V(h, f_S) - \epsilon_S(h, f_V)] + [\epsilon_V(h) + \epsilon_S(h)]) \\
&= D_{S,T,V}(f_S, f_T, f_V, h) + \frac{1}{2}[\epsilon_V(h) + \epsilon_S(h)]
\end{aligned}
$$

## D.2    PROOF OF EQ.4

$$
\begin{aligned}
\epsilon_T(h) &\leq \frac{1}{2}[\epsilon_T(f_S, f_T) + \epsilon_T(f_V, f_T) + \epsilon_T(h, f_S) + \epsilon_T(h, f_V) + \epsilon_V(f_S, f_V) + \epsilon_S(f_V, f_S) \\
&\quad - \epsilon_V(h, f_S) - \epsilon_S(h, f_V) + \epsilon_V(h) + \epsilon_S(h)] \\
&\leq \frac{1}{2}[\epsilon_T(f_S, f_S^*) + \epsilon_T(f_S^*, f_T^*) + \epsilon_T(f_T^*, f_T) + \epsilon_T(f_V, f_V^*) + \epsilon_T(f_V^*, f_T^*) + \epsilon_T(f_T^*, f_T) \\
&\quad + \epsilon_T(h, f_S^*) + \epsilon_T(f_S, f_S^*) + \epsilon_T(h, f_V^*) + \epsilon_T(f_V^*, f_V) \\
&\quad + \epsilon_V(f_S, f_S^*) + \epsilon_V(f_S^*, f_V^*) + \epsilon_V(f_V^*, f_V) + \epsilon_S(f_S, f_S^*) + \epsilon_S(f_S^*, f_V^*) + \epsilon_S(f_V^*, f_V) \\
&\quad + \epsilon_V(f_S, f_S^*) - \epsilon_V(h, f_S^*) + \epsilon_S(f_V, f_V^*) - \epsilon_S(h, f_V^*)] \\
&= \frac{1}{2}[\epsilon_V(h) + \epsilon_S(h)] + D_{S,T,V}(f_S^*, f_T^*, f_V^*, h) + \theta
\end{aligned}
$$

## D.3    CONSISTENCY

In this section, we tackle a general problem associated with the consistency between the algorithm and theory in domain adaptation. The triangle inequality is essential to build the theory and the measurement of the source error as well as terms related to the discrepancy should be the same. These requirements should be satisfied by any method that introduces an upper bound to approximate the target error. However, most of the upper bound based methods violate these rules which is known as the gap between the algorithm and theory. For instance, MCD (Saito et al., 2017) chooses cross entropy for the source error but replaces the discrepancy with a $L_1$ norm between the predictions of two classifiers. As for DANN (Ganin et al., 2016), it uses logistic loss as a surrogate to approximate 0-1 loss which no longer satisfies the triangle inequality. Despite the fact that our proposal does not serve as a perfect cure to this problem, we can prove that the proposed MCMD asymptotically satisfies the consistency.

First of all, we show that MCMD obeys the triangle inequality under the following circumstance. For the case where two hypotheses agree on the point $x$ ($y = l_{h_1}(x) = l_{h_2}(x), l_{h_3}(x) = y'$; this

condition is met when we use triangle inequality to derive the upper bound in Eq.1 except for $f_T, h$), given the definition in Eq.8:

$$\text{mcmd}(h_1, h_3; x) + \text{mcmd}(h_2, h_3; x)$$
$$= \max(|\log h_1(y|x) - \log h_3(y|x)|, |\log h_1(y'|x) - \log h_3(y'|x)|)$$
$$+ \max(|\log h_2(y|x) - \log h_3(y|x)|, |\log h_2(y'|x) - \log h_3(y'|x)|)$$
$$\geq |\log h_1(y|x) - \log h_3(y|x)| + |\log h_2(y|x) - \log h_3(y|x)|$$
$$\geq |\log h_1(y|x) - \log h_2(y|x)| = \text{mcmd}(h_1, h_2; x)$$

As the training proceed, the target error of $h$ will be minimized, which means the discrepancy between $h, f_T$ over domain $T$ is constantly reduced. Given the assumption that $f_T, h$ gradually agree on $T$, we can conclude that our proposal asymptotically satisfies the triangle inequality.

Then we prove that the cross-entropy loss is a special case of MCMD by reasonably assuming $f_S(y|x) = 1$ and $l_{f_S}(x) = l_h(x) = y$ for $(x, y) \in S$. According to Eq.8, the source error of $h$ defined based on MCMD can be written as (the same goes for $V$):

$$\epsilon_S(h) = \mathbb{E}_{x \sim S}[\text{mcmd}(h, f_S; x)] = \mathbb{E}_{x,y \sim S}|\log f_S(y|x) - \log h(y|x)|$$
$$= -\mathbb{E}_{x,y \sim S}[\log h(y|x)]$$

### D.4 INTERPRETABILITY

In this section, we explain the relation between $\boldsymbol{L_{dis}}$ and CGAN (Mirza & Osindero, 2014) to prove that the proposal can reduce the conditional discrepancy between domains. According to the constraints of hypothesis spaces (Eq.6), $f'_S$ and $f'_V$ must both classify the source ($S$) and labeled target domains ($V$). Besides, $f'_S$ tends to be more confident about the predictions on $V$ and $f'_V$ is supposed to be more confident about $S$ based on the formula of $\theta$ (Eq.5). Then we can derive that a part of $Ldis$ can be reformed as an objective of CGAN, where $f'_S, f'_V$ are two discriminators which regard $V_g, S_g$ as the real data respectively:

$$\max_{f'_S \in H_S, f'_V \in H_V} [\epsilon_{V_g}(f'_S, f'_V) + \epsilon_{S_g}(f'_S, f'_V)]$$
$$= \max_{f'_S \in H_S, f'_V \in H_V} \{\mathbb{E}_{x,y \in V_g}[\log f'_S(y|x) + \log(1 - f'_V(y|x))]$$
$$+ \mathbb{E}_{x,y \in S_g}[\log f'_V(y|x) + \log(1 - f'_S(y|x))]\}$$
$$= \max_{f'_S \in H_S} [\mathbb{E}_{x,y \in V_g} \log f'_S(y|x) + \mathbb{E}_{x,y \in S_g} \log(1 - f'_S(y|x))]$$
$$+ \max_{f'_V \in H_V} [\mathbb{E}_{x,y \in S_g} \log f'_V(y|x) + \mathbb{E}_{x,y \in V_g} \log(1 - f'_V(y|x))]$$

Then we discuss the case where two hypotheses disagree. By introducing two additional distributions $T_g^{f'_S \setminus f'_T}, T_g^{f'_T \setminus f'_S}$, we divide the target domain into two parts labeled by $f'_S$ and $f'_T$ respectively based on the difference of their prediction confidence (for simplicity, we assume $f'_S(y_s|x) \geq f'_T(y_s|x)$ and $f'_T(y_t|x) \geq f'_S(y_t|x)$):

$$\begin{cases} T_g^{f'_S \setminus f'_T} &= \{x, y_s | x \sim T_g, y_s = l_{f'_S}(x), y_t = l_{f'_T}(x) : \\ & \log f'_S(y_s|x) - \log f'_T(y_s|x) \geq \log f'_T(y_t|x) - \log f'_S(y_t|x)\} \\ T_g^{f'_T \setminus f'_S} &= \{x, y_t | x \sim T_g, y_s = l_{f'_S}(x), y_t = l_{f'_T}(x) : \\ & \log f'_T(y_t|x) - \log f'_S(y_t|x) > \log f'_S(y_s|x) - \log f'_T(y_s|x)\} \end{cases}$$

Now we can derive that a part of our objective can be reformed as an objective of CGAN, where $f'_S$ is a discriminator that regards labeled data and a part of pseudo labeled target data as the real data ($S_g \cup V_g \cup T_g^{f'_S \setminus f'_T}$). When combined with the constraint that $f'_S$ must classify $S, V$, a part of our objective w.r.t $f'_S$ becomes:

$$\max_{f'_S \in H_S} [\epsilon_{T_g}(f'_S, f'_T) - \epsilon_{S_g \cup V_g}(f'_S)] = \max_{f'_S \in H_S} [\mathbb{E}_{x,y \in S_g \cup V_g \cup T_g^{f'_S \setminus f'_T}} \log f'_S(y|x)$$
$$+ \mathbb{E}_{x,y \in T_g^{f'_T \setminus f'_S}} \log(1 - f'_S(y|x))] + const$$

Owing to the power of semi-supervised regularization, $f_T'$ is usually more confident than $f_S'$ on the unlabeled target data which gives the algorithm enough fake data to optimize. Analogously, $f_3$ can be regarded as a discriminator of CGAN that tries to align the distributions of $T_g^{f_T' \setminus f_V'}$ and $S_g \cup V_g \cup T_g^{f_V' \setminus f_T'}$.

### D.5 VALIDITY

In Sec.3.3, we make the following assumption that it is possible to build subspaces $H_S, H_T, H_V \subseteq H$ such that:

$$D_{S,T,V}(f_S, f_T, f_V, h) \leq \max_{f_S' \in H_S, f_T' \in H_T, f_V' \in H_V} D_{S,T,V}(f_S', f_T', f_V', h)$$

A sufficient condition for this would be $f_S \in H_S, f_T \in H_T, f_V \in H_V$. This condition can be easily met for $H_S, H_V$ since $S, V$ are fully labeled. As for $H_T$, the assumption is hard to prove theoretically, thus we show the validity of the inequality by experimental results instead. We choose an adaptation scenario where the domain gap is large (Product to Clipart scenario of Office-Home dataset). We use the full source and target labels to estimate $D_{S,T,V}(f_S, f_T, f_V, h)$ as the ground truth. The upper bound $D_{S,T,V}(f_S', f_T', f_V', h)$ is estimated by the maximum inside subspaces $H_S, H_T, H_V$ defined by Eq.6,7 in Sec.3.5. Fig.7 demonstrates that our proposal remains a valid upper bound in practice even if the domain gap is so large that the subspace $H_T$ we built is not likely contains $f_T$.

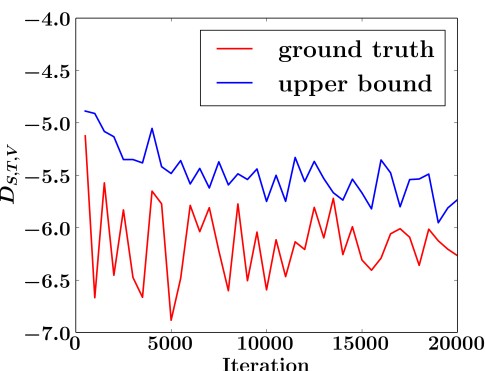

Figure 7: The estimated ground truth and upper bound of $D_{S,T,V}$ from Product to Clipart scenario in Office-Home dataset.

## E RADEMACHER COMPLEXITY

Let $H_F$ be a class of real-valued functions mapping $x \to \{0, 1\}$ and $\tilde{S} = \{x_1, ..., x_m\}$ a finite sample drawn i.i.d. according to a distribution $S$, the empirical Rademacher complexity of $H_F$ is defined as follows:

$$\hat{\Re}_{\tilde{S}}(H_F) = \frac{2}{m} \mathbb{E}_\sigma \left[ \sup_{h \in H_F} | \sum_{i=1}^m \sigma_i h(x_i) | \right]$$

The expectation is taken over $\sigma = (\sigma_1, ..., \sigma_n)$ where $\sigma_i$ is an independent uniform random variable taking values in $\{-1, +1\}$. Following the established theory proposed by Mansour et al. (2009), we denote the empirical average risk of a hypothesis $h : x \to \{0, 1\}$ by $\hat{R}_{\tilde{S}}(h)$ and its expectation over samples drawn according to the distribution by $R_S(h)$. According to Koltchinskii & Panchenko (2000); Bartlett & Mendelson (2002), for any $\sigma > 0$, with probability at least $1 - \sigma$ over samples $\tilde{S}$ of size $m$, the following inequality holds for all $h \in H_F$:

$$R_S(h) \leq \hat{R}_{\tilde{S}}(h) + \hat{\Re}_{\tilde{S}}(H_F) + 3\sqrt{\frac{\log \frac{2}{\sigma}}{2m}}$$

Based on the above theory, assuming the loss function $\epsilon$ is bounded with $M$, we can scale the loss to $[0, 1]$ and bound it with:

$$\frac{\epsilon_S(f, f')}{M} \leq \frac{\hat{\epsilon}_{\tilde{S}}(f, f')}{M} + \hat{\Re}_{\tilde{S}}(H^\epsilon_{A,B}/M) + 3\sqrt{\frac{\log \frac{2}{\sigma}}{2m}}$$

where $H^\epsilon_{A,B}$ represents a new space of hypotheses mapping $x \to \{\epsilon(f(x), f'(x))|f \in H_A, f' \in H_B, \epsilon \leq M\}$.

Recall Eq.1 that the target error is upper bounded by:

$$\epsilon_T(h) \leq \frac{1}{2}[\epsilon_S(h) + \epsilon_V(h)] + D_{S,T,V}(f_S, f_T, f_V, h)$$

where $D_{S,T,V}(f_S, f_T, f_V, h)$ is bounded by:

$$D_{S,T,V}(f_S,f_T,f_V,h) \leq \sup\nolimits_{f'_S \in H_S, f'_T \in H_T, f'_V \in H_V} \tfrac{1}{2}[\epsilon_T(f'_S, f'_T) + \epsilon_T(f'_T, f'_V) + \epsilon_T(h, f'_S) + \epsilon_T(h, f'_V) + \epsilon_V(f'_S, f'_V) + \epsilon_S(f'_S, f'_V) - \epsilon_V(h, f'_S) - \epsilon_S(h, f'_V)]$$

By applying:

$$\begin{cases} \epsilon_S(f'_S, f'_V) \leq \epsilon_S(f'_S, h) + \epsilon_S(f'_V, h) \\ \epsilon_V(f'_S, f'_V) \leq \epsilon_V(f'_S, h) + \epsilon_V(f'_V, h) \end{cases}$$

we can further upper bound the $D_{S,T,V}(f_S, f_T, f_V, h)$ with:

$$d_{S,T,V}(f'_S, f'_T, f'_V, h) = \sup\nolimits_{f'_S \in H_S, f'_T \in H_T, f'_V \in H_V} \tfrac{1}{2}[\epsilon_T(f'_S, f'_T) + \epsilon_T(f'_T, f'_V) + \epsilon_T(h, f'_S) + \epsilon_T(h, f'_V) + \epsilon_V(h, f'_V) + \epsilon_S(f'_S, h)]$$

Now let $\tilde{S}$ be a random sample of size $m$ from domain $S$, let $\tilde{T}$ be a random sample of size $n$ from domain $T$ and let $\tilde{V}$ be a random sample of size $l$ from domain $V$, for any $\sigma > 0$, with probability at least $1 - \sigma$, the following inequality holds for all $h \in H_F$:

$$d_{S,T,V}(f'_S, f'_T, f'_V, h)$$

$$\leq \sup_{f'_S \in H_S, f'_T \in H_T, f'_V \in H_V} \frac{1}{2}\{[\hat{\epsilon}_{\tilde{T}}(f'_S, f'_T) + \hat{\epsilon}_{\tilde{T}}(f'_T, f'_V) + \hat{\epsilon}_{\tilde{T}}(h, f'_S) + \hat{\epsilon}_{\tilde{T}}(h, f'_V) + \hat{\epsilon}_{\tilde{V}}(h, f'_V) + \hat{\epsilon}_{\tilde{S}}(f'_S, h)]$$

$$+ \hat{\Re}_{\tilde{T}}(H^\epsilon_{S,T}) + \hat{\Re}_{\tilde{T}}(H^\epsilon_{T,V}) + \hat{\Re}_{\tilde{T}}(H^\epsilon_{F,S}) + \hat{\Re}_{\tilde{T}}(H^\epsilon_{F,V}) + \hat{\Re}_{\tilde{S}}(H^\epsilon_{F,S}) + \hat{\Re}_{\tilde{V}}(H^\epsilon_{F,V})$$

$$+ 3M(\sqrt{\frac{\log \frac{2}{\sigma}}{2m}} + \sqrt{\frac{\log \frac{2}{\sigma}}{2l}} + 4\sqrt{\frac{\log \frac{2}{\sigma}}{2n}})\}$$

