# OpenReview forum: "Semi-supervised Domain Adaptation via Joint Error based Triplet Alignment"
_ICLR.cc/2024/Conference — ICLR 2024 Conference Withdrawn Submission_

### Official Review · Reviewer_XYg3 · 2023-10-31

**Soundness:** 3 good
**Presentation:** 1 poor
**Contribution:** 2 fair
**Rating:** 3
**Confidence:** 4

**Summary:**

The paper presents a new framework for semi-supervised domain adaptation (SSDA) that establishes an upper bound on target error. This framework introduces a method called Joint Error-based Triplet Alignment (JTA), which performs alignments not only between the labeled source domain and the unlabeled target domain but also between the labeled source domain and the labeled target domain. As a result, their empirical studies demonstrate that JTA can reduce domain gaps and enhance feature learning by explicitly considering the alignment for the labeled target data. The paper also introduces a dissimilarity metric known as Maximum Cross Margin Discrepancy (MCMD) to bridge the gap between theory and algorithm, ensuring the consistency of the target error bound. The main problem of this paper is the lack of sufficient details to understand and follow their motivation and derivation. Given the promising empirical results presented in the paper, I strongly recommend that the authors consider a complete rewrite of the paper, focusing on delivering a clear and well-motivated presentation. This should involve providing comprehensive derivations with sufficient details or citations, ensuring that each step of each equation is transparently explained for the benefit of the reader's understanding.

**Strengths:**

The performance of the proposed work is promising.

**Weaknesses:**

1. I find the paper's motivation unclear. To be specific, the upper bound of the hypothesis regarding the unlabeled target domain should be the most crucial starting point for readers to comprehend what the proposed method aims to address. However, the lack of an explanation for the proof of Equation (1) makes it extremely difficult for me to grasp and follow. Concerning D.1, I am unsure how the first equation of the unlabeled target error bound was derived. If it stems from Ben David's theorem (assuming my recollection is accurate, Ben David did not derive any error bound under semi-supervised settings) or the work of others, it would be beneficial to provide citations so that readers can fully contextualize and understand the subject matter.

2. What is the source of the intractability, particularly for f_{S} and f_{V}? Given that both S and V are fully labeled, it seems reasonable to assume that a straightforward optimization approach like empirical risk minimization (ERM) could yield a reasonable approximation for f_{S} and f_{V). The mention of intractability is often made within the framework of variational inference, where certain integrations cannot be feasibly solved. Providing a clear explanation of this intractability would significantly enhance the paper's motivation.

3. How is the reduction of the error term achieved between two fixed true labeling functions? I want to emphasize that "true" here means unchanging or fixed. The paper is proving a complex upper bound derivation, and its clarity is hindered by inconsistent definitions throughout, making it difficult to follow.

4. The t-SNE visualization, without any indications of the class labels for each data sample, fails to convey meaningful information. In fact, I find the t-SNE visualization rather perplexing. I recommend that the authors consider sharing the code for their implementation with the reviewers. This would serve not only to confirm the reproducibility of their work but also to enhance the reviewers' understanding of the proposed methodology.

5. The experimental setup lacks clarity, particularly in the context of semi-supervised domain adaptation, where the number of labeled target samples and the way to select the labeled target sample are crucial. It is important to provide sufficient details regarding the sample selection process.

6. The authors assert that [1] violates the triangle inequality without providing a thorough explanation or derivation. This is a strong claim, as it implies [1] is a departure from well-established theoretical foundations, especially considering that [1] is published on a top tire. To support their claim, the authors should conduct in-depth elaboration and studies.

### Reference

[1] Yuchen Zhang, Tianle Liu, Mingsheng Long, and Michael Jordan. Bridging theory and algorithm for domain adaptation. In Proceedings of the 36th International Conference on Machine Learning, volume 97, pp. 7404–7413. PMLR, 2019.

**Questions:**

1. Could you please clarify what is meant by the conditional distribution referred to in Section 3.1? To be specific, which random variables are conditioned on which other random variables? Based on the authors’ preliminary at the beginning of the section that both f_{S} and f_{V} are true labeling functions (true means fixed and deterministic). Meanwhile, I am confused by the idea of describing a mapping function (mapping function is normally deterministic) as a distribution (sampling from a distribution is stochastic). How come a stochastic term can be used to describe a deterministic notation? Can you elaborate on this?

2. To me, the loss introduced in this work appears to be an extension of the one (MDD) presented in [1] to the semi-supervised setting. I would appreciate it if the authors could offer a comprehensive discussion outlining the primary distinctions between [1] and their proposed approach, excluding the consideration of the semi-supervised setting and the violation of the triangle inequality.

### Reference

[1] Yuchen Zhang, Tianle Liu, Mingsheng Long, and Michael Jordan. Bridging theory and algorithm for domain adaptation. In Proceedings of the 36th International Conference on Machine Learning, volume 97, pp. 7404–7413. PMLR, 2019.

---

### Official Review · Reviewer_tePx · 2023-10-31

**Soundness:** 2 fair
**Presentation:** 2 fair
**Contribution:** 1 poor
**Rating:** 3
**Confidence:** 3

**Summary:**

The paper at hand proposes a method for domain adaptation by including some labeled data from the target domain. A "triplet alignment" is introduce which aims for aligning feature distributions as well as minimizing classification error.

**Strengths:**

+ relevant problem

**Weaknesses:**

- The paper is quite hard to read and understand. Figures are rather small. Honesty speaking Fig. 1 even confused me more than it helped me to understand the approach.
- Experimental results are hard to interpret and judge. If I read it correctly, the effect of data augmentation seems significant. When comparing without data augmentation  (ours* in Tab. 1) the advantages over previously proposes approaches seems marginal (if at all). I also miss confidence intervals.

**Questions:**

- What are clear advantages of the approach -- e.g., the claim that "data augmentation is not nessaccary for our approach" (besides still having a significant impact) is not well motivated.
- What are limitation of the approach?

---

### Official Review · Reviewer_hsiV · 2023-11-01

**Soundness:** 3 good
**Presentation:** 3 good
**Contribution:** 1 poor
**Rating:** 1
**Confidence:** 5

**Summary:**

This paper proposed a joint error based triplet alignment approach to solve the semi-supervised domain adaptation problem. They evaluated on several cross-domain benchmarks by comparing with several methods. Generally, the paper is easy to follow. However, the novelty is not enough.

**Strengths:**

This paper proposed a joint error based triplet alignment approach to solve the semi-supervised domain adaptation problem. They evaluated on several cross-domain benchmarks by comparing with several methods. Generally, the paper is easy to follow. They show various results to examine their methods.

**Weaknesses:**

The novelty is not enough. The joint error based triplet alignment is not new, which is an extension of maximum cross margin discrepancy to three subsets, source, labeled target and unlabeled target. Eventual model is also very complicated.

The model performance is not good enough. Especially compared with DECOTA in Table 1 & 2, it is very comparable. Also for semi-supervised setting, the selected target samples are very essential. There is no standard variance. Also t-test is needed to examine the significance.

**Questions:**

The clarification of model novelty.
The performance improvement.

---

### Official Review · Reviewer_TnGf · 2023-11-02

**Soundness:** 3 good
**Presentation:** 3 good
**Contribution:** 2 fair
**Rating:** 3
**Confidence:** 3

**Summary:**

This work introduces a Triplet Alignment approach for semi-supervised domain adaptation. It simultaneously minimizes the joint error among different domains and the error rate on labeled data.

**Strengths:**

1.	The motivation for this work is clear. It aims to address the challenge of semi-supervised domain adaptation, particularly when only a limited number of annotated examples are available in the target domain. The proposed method optimizes both the classification loss and the joint error across source, labeled, and unlabeled target domains simultaneously.
2.	The proposed models are presented in a clear and comprehensible manner.

**Weaknesses:**

1.	The proposed model, to the best of my knowledge, lacks significant novelty as it closely resembles the approach in [2]. It would be helpful to explicitly identify the main difference.
2.	The choice of baseline methods in this work appears to be less competitive. Given the recent progress in semi-supervised domain adaptation (SSDA), including [1][2], it is advisable to compare the proposed method with these contemporary approaches. Furthermore, while the use of t-SNE for feature space visualization is commendable, the comparisons are made with older methods like ENT (Grandvalet & Bengio, 2005), MJE (Zhang & Harada, 2019), and MME (Saito et al., 2019). It is imperative to include comparisons with more recent methods to provide a comprehensive evaluation.
[1]  Yu, Yu-Chu, and Hsuan-Tien Lin. "Semi-Supervised Domain Adaptation with Source Label Adaptation." Proceedings of the IEEE/CVF Conference on Computer Vision and Pattern Recognition. 2023.
[2] Rahman, Md Mahmudur, Rameswar Panda, and Mohammad Arif Ul Alam. "Semi-Supervised Domain Adaptation with Auto-Encoder via Simultaneous Learning." Proceedings of the IEEE/CVF Winter Conference on Applications of Computer Vision. 2023.

**Questions:**

Please see "Weaknesses"